# Serum Concentration of Inflammatory Cytokines in Dogs with Suspected Acute Pancreatitis

**DOI:** 10.3390/vetsci8030051

**Published:** 2021-03-18

**Authors:** Soon-Won Choi, Yoon-Hwan Kim, Min Soo Kang, Yunho Jeong, Jin-Ok Ahn, Jung Hoon Choi, Jin-Young Chung

**Affiliations:** 1Department of Veterinary Internal Medicine and Institute of Veterinary Science, College of Veterinary Medicine, Kangwon National University, Chuncheon-si 24341, Korea; linevet@naver.com (S.-W.C.); luperskim@gmail.com (Y.-H.K.); jsteve35@outlook.com (Y.J.); joahn@kangwon.ac.kr (J.-O.A.); 2Department of Veterinary Anatomy and Institute of Veterinary Science, College of Veterinary Medicine, Kangwon National University, Chuncheon-si 24341, Korea; imkangms@kangwon.ac.kr (M.S.K.); jhchoi@kangwon.ac.kr (J.H.C.)

**Keywords:** acute pancreatitis, antibody array, cytokine, dog, enzyme-linked immunosorbent assay

## Abstract

Acute pancreatitis is an acute inflammatory process in the pancreas that is common in dogs. This study was designed to compare cytokines between healthy dogs and dogs with suspected acute pancreatitis. For the canine cytokine antibody array, three healthy dogs and three dogs with suspected acute pancreatitis were included. Interleukin (IL)-2, IL-6, IL-10, GM-CSF, and TNF-α were not detected in either group based on the results. Conversely, IL-8 (*p* = 0.035), Monocyte Chemoattractant Protein-1 (MCP)-1 (*p* = 0.0138), Receptor for Advanced Glycation Endproducts (RAGE) (*p* = 0.0079), and stem cell factor (SCF) (*p* = 0.034) were significantly increased in dogs with suspected acute pancreatitis. However, vascular endothelial growth factor (VEGF) (*p* = 0.6971) did not differ significantly between groups. For the canine serum Enzyme-Linked Immunosorbent Assay (ELISA), eight healthy dogs and eight dogs with suspected acute pancreatitis were included. ELISA revealed that IL-8 (*p* < 0.0001), MCP-1 (*p* < 0.0001), RAGE (*p* = 0.006), and SCF (*p* = 0.0002) were all significantly upregulated in the experimental group. We confirmed multiple patterns of cytokines in suspected acute pancreatitis of dogs via canine cytokine antibody array using a small quantity of serum. After this procedure, we reevaluated the cytokines, which were significantly increased in dogs with suspected acute pancreatitis, by ELISA, with more samples. Through this study, we confirmed that MCP-1, RAGE, and SCF were newly suggested factors in dogs with suspected acute pancreatitis.

## 1. Introduction

Pancreatitis is the most common disease of the exocrine pancreas in dogs [1]. The clinical signs of pancreatitis in dogs range widely and include vomiting, diarrhea, and dehydration, although many cases have subclinical or mild nonspecific clinical signs [2]. Therefore, it is sometimes difficult to diagnose pancreatitis while the dog is still alive. In one study, 8% of 200 dogs that had died or had been euthanized for various reasons showed macroscopic evidence of pancreatitis at necropsy [3].

Pancreatitis is divided into acute and chronic pancreatitis. Acute pancreatitis is an acute inflammatory process in the pancreas with the histologic presence of edema, neutrophilic infiltration, and necrosis that usually occurs with various degrees of involvement in other tissues [4]. Chronic pancreatitis is characterized by continuous inflammation with irreversible changes. Approximately 50% of dogs diagnosed with pancreatitis on necropsy showed evidence of chronic pancreatitis and 30% had lesions suggestive of acute pancreatitis [5]. Acute and chronic pancreatitis can only be differentiated based on histopathology. Unfortunately, surgical biopsy of the pancreas is invasive and requires anesthesia, which may be high-risk to the patients. So surgical biopsy of pancreas is not routinely suggested for the patients [5]. Chronic pancreatitis usually shows mild nonspecific clinical signs, while acute pancreatitis can develop into a severe form that can lead to systemic inflammatory response syndrome [3,6]. A few studies have shown that the mortality rate for acute pancreatitis is high, ranging from 27 to 58% [2,7].

For the diagnosis of pancreatitis, abdominal ultrasound imaging, general clinical pathology, serum amylase activity, total serum lipase activity, trypsin-like immunoreactivity, and pancreatic lipase immunoreactivity (PLI) are usually considered. However, most of these diagnostic tools are not specific to the diagnosis of pancreatitis in dogs. Moreover, serum amylase and lipase were believed to be key factors for the diagnosis of pancreatitis; however, their levels can be abnormal in response to other disorders and are therefore no longer considered key factors in diagnosing pancreatitis [5,8]. Pancreatic lipase immunoreactivity is used for the detection of pancreatic lipase, which is only released by pancreatic acinar cells. Currently, this is the only specific assay for detecting pancreatic lipase [9].

Abdominal ultrasound imaging is commonly used in the diagnostic evaluation of canine acute pancreatitis, and it is also used to rule out other diseases that may cause similar clinical signs [10].

Although pancreatitis is common in veterinary and human medicine, the mechanisms of the disease are not fully understood. However, complex interactions exist between cytokines, inflammation, and the adaptive responses in the body. It has been shown that inflammatory cytokines play important roles in the progression and severity of pancreatitis [11]. This study suggested that new cytokines, Monocyte Chemoattractant Protein-1 (MCP)-1, Receptor for Advanced Glycation Endproducts (RAGE), and stem cell factor (SCF), which could be involved in the pathogenesis of suspected acute pancreatitis in dogs, differed from those in normal controls.

## 2. Materials and Methods

### 2.1. Case Selection

A total of 22 dogs were enrolled in this study, among which, 6 were included in the cytokine antibody array analysis and the other 16 dogs were included in ELISA analysis. Of the six dogs in the cytokine antibody array, three were healthy and the other three suffered from acute pancreatitis. Of the 16 dogs evaluated by ELISA analysis, 8 were healthy dogs and the other 8 suffered from suspected acute pancreatitis.

Dogs with suspected acute pancreatitis were included, while those that had other diseases before diagnosis or that had been hospitalized for >3 days at another hospital were excluded. 

The diagnosis of acute pancreatitis was established based on a positive SNAP cPL test and morphologic evidence of pancreatitis was obtained by ultrasonography. All of the diagnosed dogs with acute pancreatitis suffered from clinical signs including anorexia, vomiting, and abdominal pain. However, we could not perform histopathological analysis. Therefore, the patients were named as dogs with suspected acute pancreatitis.

### 2.2. Canine Cytokine Antibody Array

Serum was obtained from all dogs, and 100 µL were used for the array protocol. Serum was diluted 1:10 and probed for cytokines using a RayBiotech Canine cytokine antibody array kit (catalog number QAC-CYT-1, RayBiotech Inc., Norcross, GA, USA).

Briefly, membranes were blocked with blocking buffer, after which the serum was individually added and incubated at 4 °C overnight. The membranes were then washed, after which 80 µL of biotinylated antibody cocktail was added to each well and the samples were incubated at room temperature for 2 h. The membranes were then washed, after which 80 µL of Cy3 equivalent dye-conjugated streptavidin was added to each well and the samples were incubated at room temperature for 1 h and 30 min in the dark. The membranes were then washed.

Slides were scanned using a GenePix 4100A scanner (Axon Instrument, Sunnyvale, CA, USA). The slides were completely dried before scanning and scanned within 24–48 h. Slides were scanned at 10 µm resolution, optimal laser power and PMT. After the images of the scans were obtained, they were gridded and quantified with the GenePix software (Axon Instrument, Sunnyvale, CA, USA) and analyzed according to the recommendations from RayBiotech. Data were imported into an Excel spreadsheet and normalized against a control across membranes, after which final values were calculated using the RayBio canine cytokine 1 analysis tool.

Antibodies of interleukin (IL)-2, IL-6, IL-8, granulocyte–macrophage colony-stimulating factor (GM-CSF), MCP-1, RAGE, SCF, tumor necrosis factor- α (TNF-α), and vascular endothelial growth factor (VEGF) were analyzed with a canine cytokine antibody array.

### 2.3. Canine Serum Enzyme-Linked Immunosorbent Assay

Serum cytokines (IL-8, MCP-1, RAGE, SCF) were quantified in healthy dogs and dogs with suspected acute pancreatitis by ELISA according to the manufacturer’s instructions (RayBiotech Inc., Norcross, GA, USA). Briefly, 50 µL of dog diluted serum (IL-8, MCP-1, RAGE, SCF; 1:20, 1:4, 1:4, 1:4) was added to each well and incubated according to the manufacturer’s instructions. Upon completion of the assay procedure, the plate was read at 450 nm using a VersaMax ELISA microplate reader (Molecular Devices, San Jose, CA, USA).

### 2.4. Statistical Analysis

For the analysis of the cytokine antibody array and the ELISA data, values are expressed as the means ± Standard Deviation (SD). Healthy dogs and dogs with suspected acute pancreatitis were compared by a t-test. A *p*-value < 0.05 was taken to indicate significance, and SPSS10.0 (SPSS Inc, Chicago, IL, USA) was used for statistical analysis.

## 3. Results

### 3.1. Canine Cytokine Antibody Array

Three healthy dogs and three dogs with suspected acute pancreatitis were included. The dogs with suspected acute pancreatitis included one poodle, one Pomeranian, and one mixed dog. Additionally, two were castrated males and one was a spayed female. The mean age of the suspected acute pancreatitis dogs was 11.7 years. The healthy dogs included one beagle, one poodle, and one mixed dog, two of which were castrated males and one that was a spayed female. The mean age of the healthy dogs was 6.7 years. Cytokine microarray analysis of the serum samples collected in both groups revealed that IL-2, IL-6, IL-10, GM-CSF, and TNF-α were not present in either group. On the other hand, IL-8 (*p* = 0.035), MCP-1 (*p* = 0.0138), RAGE (*p* = 0.0079), and SCF (*p* = 0.034) were significantly increased in the dogs with suspected acute pancreatitis (Figure 1). However, VEGF (*p* = 0.6971) did not differ significantly between groups.

### 3.2. Canine Serum Enzyme-Linked Immunosorbent Assay

Eight healthy dogs and eight dogs with suspected acute pancreatitis were analyzed. The dogs with suspected acute pancreatitis included one poodle, one beagle, one cocker spaniel, one Yorkshire terrier, one Maltese, one spitz, one miniature pinscher, and one mixed dog, three of which were castrated males and five that were spayed females. The mean age of the suspected acute pancreatitis dogs was 10.5 years. The healthy dogs included six beagles, one poodle, and one mixed dog, three of which were castrated males, one that was a spayed female, and four that were female. The mean age of the healthy dogs was 5.9 years. The ELISA analysis of serum samples collected from both groups revealed that IL-8 (*p* < 0.0001), MCP-1 (*p* < 0.0001), RAGE (*p* = 0.006), and SCF (*p* = 0.0002) were all significantly increased in dogs with suspected acute pancreatitis (Figure 2).

## 4. Discussion

Acute pancreatitis can lead to systemic inflammatory response syndrome. Systemic inflammatory response syndrome refers to the clinical signs of a complex physiologic response to a nonspecific insult of either infectious or noninfectious origin. For the diagnosis of systemic inflammatory response syndrome, history, physical examination, blood test, urinalysis, and diagnostic imaging should be performed in any critically ill patient [3,5,6]. In this study, all the dogs that had other diseases before diagnosis or that had been hospitalized for >3 days at another hospital were excluded. So we excluded dogs with systemic inflammatory response syndrome.

Pancreatitis develops in response to excessive activation of trypsin and other pancreatic proteases within the pancreas. This activation may be caused by oxidative stressors or hypotension. After the initial activation of pancreatic enzymes, local inflammation should be followed [12]. Many studies have been conducted to elucidate the pathophysiology of acute pancreatitis. Unfortunately, even with these efforts, the mechanism is not fully understood [7]. This report is one effort to elucidate the pathophysiology of suspected acute pancreatitis in dogs by suggesting new factors: MCP-1, RAGE, and SCF.

Many cytokines are activated before, during, and after inflammatory response occurs within the pancreas via a complex phenomenon known as a “cytokine storm”. Cytokines are small proteins produced by stimulation that up-regulate or down-regulate many aspects of the inflammatory process. Cytokines by T cells can lead to T helper cell type (Th)1 or Th2 immune responses, which mediate cellular immunity and humoral immunity, respectively. Each cytokine can act synergistically with others, and the functions of these cytokines are largely overlapping [13].

In this study, cytokine microarray analysis of the serum samples collected in both groups revealed that IL-2, IL-6, IL-10, GM-CSF, and TNF-α were not detected in either group. On the other hand, IL-8, MCP-1, RAGE, and SCF were significantly increased in the dogs with suspected acute pancreatitis. However, VEGF did not differ significantly between groups. After this scanning procedure, we re-evaluated the cytokines, which were significantly increased in dogs with suspected acute pancreatitis, using ELISA analysis. Through this study, we confirmed that IL-8, MCP-1, RAGE, and SCF were all significantly increased in dogs with suspected acute pancreatitis.

IL-8 is known to be an initiating cytokine that drives neutrophil migration early in many inflammatory diseases, including acute pancreatitis. During systemic inflammatory conditions such as sepsis or systemic inflammatory response syndrome, morbidity and mortality can be predicted based on elevated levels of cytokines. When compared with TNF-α and IL-1β, injection of IL-8 does not induce a shock-like state. A previous study confirmed that levels of IL-8 were elevated in acute pancreatitis and correlated with levels of the neutrophil elastase, a marker of neutrophil activation [14]. In the present study, the levels of IL-8 were also found to be significantly elevated in dogs with suspected acute pancreatitis upon canine cytokine antibody array (*p* = 0.035) and canine cytokine ELISA (*p* < 0.0001).

Only a few studies have investigated MCP-1, RAGE, and SCF in veterinary medicine, especially as they relate to pancreatic problems. One study compared the serum concentration of MCP-1 in healthy and critically ill dogs. MCP-1 is a primary regulator of monocyte mobilization from bone marrow that increases inflammatory disorders in critically ill people. In their study, sepsis, parvovirus gastroenteritis, immune mediated hemorrhagic anemia, and trauma were included in the critically ill dog group, and the results confirmed that the levels of MCP-1 in critically ill dogs were significantly increased when compared with healthy dogs [15]. RAGE is a receptor of the immunoglobulin super-family that is expressed in various cells implicated in the immune-inflammatory response. Some studies have shown that ligand-RAGE interaction on cells mediates cell migration and the up-regulation of pro-inflammatory cells [16]. In this study, the levels of MCP-1 and RAGE were also found to be significantly elevated in dogs with suspected acute pancreatitis by canine cytokine antibody array and canine cytokine ELISA.

SCF is a cytokine that binds to the c-Kit receptor and is expressed by various structural and inflammatory cells. In in vitro studies, SCF was found to induce histamine, pro-inflammatory cytokines, and chemokines [17]. However, few studies have investigated SCF in the pancreas. One group showed a relationship between SCF and chronic pancreatitis. This group confirmed that not only was the total number of mast cells significantly higher in chronic pancreatitis than in a normal pancreas, but that SCF- and c-kit immunoreactive mast cells were mostly localized in fibrous tissue and around the regenerating duct [18]. However, no studies have investigated the relationship between acute pancreatitis and SCF in animal or human medicine. In the present study, the levels of SCF were found to be significantly higher in dogs with suspected acute pancreatitis by canine cytokine antibody array (*p* = 0.034) and canine ELISA (*p* = 0.0002).

IL-2 has long been recognized as central to normal immunologic function, including protection against infection. The role of IL-2 in burns and trauma has been actively stud-ied; however, there has been little effort to investigate the role of IL-2 in acute pancreatitis. One study confirmed that the level of IL-2 decreased in murine acute pancreatitis. In that study, it was assumed that IL-2 had the potential for immunomodulation [19]. GM-CSF is an immune modulatory cytokine produced by different cells that can be regulated by cytokines, antigens, or other inflammatory agents. GM-CSF has been investigated in clinical trials for its immune-modulatory effects [20]. IL-10 is an anti-inflammatory cytokine believed to inhibit the release of pro-inflammatory cytokines to prevent subsequent tissue damage. IL-10 is thought to play a protective role in acute pancreatitis [14]. In this study, IL-2, GM-CSF, and IL-10 were not detected by the canine cytokine antibody array in either group, which is concordant with the results of previous studies. Based on these results, we assumed that there was not an anti-inflammatory response.

TNF-α, which is derived predominantly from activated macrophages, is one of the major mediators of shock [14]. IL-6 is produced by various cells in response to stimulation by other cytokines, including TNF-α [14]. High levels of IL-6 have been reported in acute conditions such as sepsis, and it is an excellent predictor of disease severity that mediates acute phase response. In this study, TNF-α and IL-6 were not detected by canine cytokine antibody array. None of the dogs in this study suffered from sepsis; therefore, we can assume that the levels of TNF-α and IL-6 do not increase in pancreatitis without septic conditions.

The traditional method for cytokine detection is through the use of an ELISA. While this traditional method works well for a single protein, the overall procedure is time consuming and requires a relatively high volume of sample. Thus, conservation of precious small sample quantities becomes a challenging task. Cytokine antibody arrays, which were used in this study, are based on the multiplexed sandwich ELISA-based quantitative array platform and enable the determining of the concentration of multiple cytokines simultaneously. It combines the advantages of the high detection sensitivity and specificity of ELISA and the high throughput of arrays. For this reason, we tried to scan nine different kinds of inflammatory cytokines simultaneously in acute pancreatitis of dogs via canine cytokine antibody array using a small quantity of serum. After which we reevaluated the cytokines, which were significantly increased in the dogs with suspected acute pancreatitis, using ELISA with more samples [21].

## 5. Conclusions

In the present study, we conducted canine ELISA based on the results of a canine cytokine antibody array. Although the canine cytokine antibody array is useful for scanning for nine different kinds of inflammatory cytokines, it is expensive. Therefore, we used this tool to scan for the nine cytokines, after which IL-8, MCP-1, RAGE, and SCF were selected for the ELISA analysis based on the results of the canine cytokine antibody array. Based on the results from both assays, MCP-1, RAGE, and SCF are newly suggested factors for the identification of suspected acute pancreatitis in dogs.

## Figures and Tables

**Figure 1 vetsci-08-00051-f001:**
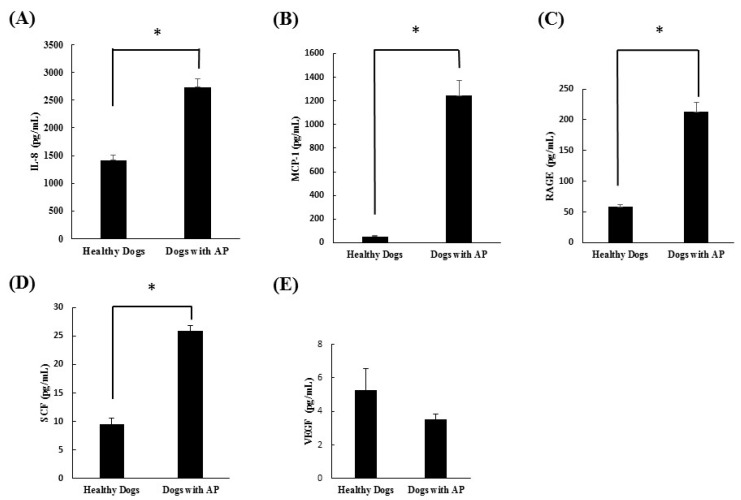
Comparison of (**A**) IL-8, (**B**) MCP-1, (**C**) RAGE, (**D**) SCF, and (**E**) VEGF in dogs with suspected acute pancreatitis (n = 3) and healthy dogs (n = 3) by canine cytokine antibody array. * *p* < 0.05. AP: suspected acute pancreatitis

**Figure 2 vetsci-08-00051-f002:**
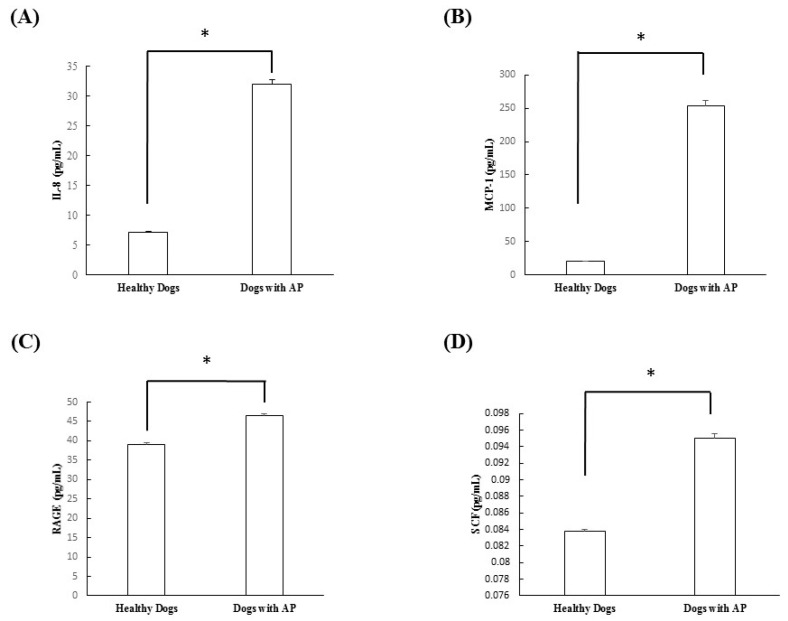
Comparison of (**A**) IL-8, (**B**) MCP-1, (**C**) RAGE, and (**D**) SCF in dogs with suspected acute pancreatitis (n = 8) and healthy dogs (n = 8) by canine serum enzyme-linked immunosorbent assay. * *p* < 0.05. AP: suspected acute pancreatitis

## Data Availability

The datasets used and/or analyzed during the current study are available from the corresponding author upon reasonable request.

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
