# Peer review of "Serum Concentration of Inflammatory Cytokines in Dogs with Suspected Acute Pancreatitis"

_vetsci, 2021, doi:10.3390/vetsci8030051_

Round 1

Reviewer 1 Report

The authors used two methods to quantify cytokines in the serum of dogs with acute pancreatitis.

Some points must be better addressed in the text of the paper: What is the objective of the research? Establish new diagnostic markers? Explain the mechanism of tissue or systemic inflammation? Or compare two methods of detecting cytokines in the serum?

Although the results are interesting, and demonstrate a difference between the cytokine profile assessed in the groups with and without pancreatitis using both methods, it is not possible to relate these cytokines to the degree of tissue damage without a histopathological examination of the pancreas.

The increase in these cytokines cannot be attributed only to the inflammatory lesion of the pancreas, as there must be some degree of peritonitis as well. In fact, as was commented by the authors, there is the activation of a systemic inflammatory process. The results showed that cytokines IL-8, RAGE, MCP-1 and SCF are increased and in my point of view it may contribute to explain the pathophysiology of systemic inflammation.

This fact needs to be better clarified and discussed by the authors in the text.

 Other few  points I would like to highlight on the text:

Abstract

Please explain here why two methods were used to quantify serum cytokines. What was the goal of it?

Lines 18-19 -This study demonstrated that serum cytokines in dogs with pancreatitis differed from those in normal controls. Therefore, these cytokines may be involved in the pathogenesis of systemic inflammation (?).

The conclusion of the abstract is different from the conclusion at the end of the work

Discussion

 Line 151 This report is one effort to elucidate the pathophysiology of acute pancreatitis in dogs.

The objective needs to be more specific and determined at the beginning of the paper

My suggestion is to discuss first the results of cytokines that increased and then those that did not show up.

 Conclusion

Line 119-220 - Based on the results from both assays, MCP-1, RAGE and SCF are newly suggested factors for identification of acute pancreatitis in dogs.

Was this final conclusion the purpose of the work?

There is some recent bibliography on pancreatitis in dogs that can be cited in the paper

Author Response

The authors used two methods to quantify cytokines in the serum of dogs with acute pancreatitis.

Some points must be better addressed in the text of the paper:

Point 1: What is the objective of the research? Establish new diagnostic markers? Explain the mechanism of tissue or systemic inflammation? Or compare two methods of detecting cytokines in the serum?

Response: With this study, we confirmed that MCP-1, RAGE and SCF were newly suggested factors in dogs with acute pancreatitis. This report is one effort to elucidate the pathophysiology of acute pancreatitis in dogs with newly suggested cytokines. We indicated about this in the abstract, introduction, discussion and conclusion part (Line 32 - 33, 74 - 76, 174 – 175, 259 - 265).

Point 2: Although the results are interesting, and demonstrate a difference between the cytokine profile assessed in the groups with and without pancreatitis using both methods, it is not possible to relate these cytokines to the degree of tissue damage without a histopathological examination of the pancreas.

Response: Acute and chronic pancreatitis can only be differentiated based on histopathology. However, surgical biopsy of pancreas is invasive and requires anesthesia, which may be high risk to the patients. So surgical biopsy of pancreas is not routinely suggested to the patients. We indicated this in the manuscript (Line 50-52) So usually in clinic, the patients, which had a positive SNAP cPL test and morphologic evidence of pancreatitis obtained by ultrasonography, were considered as patients with acute pancreatitis and treated based on acute pancreatitis. As your comment we could not confirm acute pancreatitis without histological analysis so we named this as suspected acute pancreatitis. We corrected all ‘acute pancreatitis’ to ‘suspected acute pancreatitis’. Please consider all about these.

Point 3: The increase in these cytokines cannot be attributed only to the inflammatory lesion of the pancreas, as there must be some degree of peritonitis as well. In fact, as was commented by the authors, there is the activation of a systemic inflammatory process. The results showed that cytokines IL-8, RAGE, MCP-1 and SCF are increased and in my point of view it may contribute to explain the pathophysiology of systemic inflammation. This fact needs to be better clarified and discussed by the authors in the text.

Response: In this study, all the dogs that had other diseases before diagnosis or had been hospital-ized for >3days at another hospital were excluded. So we excluded dogs with systemic inflammatory response syndrome. As your comment, we described this in the discussion part. (Line 161-168).

 Other few points I would like to highlight on the text:

Abstract

Point 1: Please explain here why two methods were used to quantify serum cytokines. What was the goal of it?

Response: We used canine cytokine antibody array for the scanning nine different kinds of inflammatory cytokines with a small quantity of serum. Unfortunately, it is expensive. So we conducted canine ELISA based on the results of a canine cytokine antibody array. We described this in the abstract part (Line 28-33).

Point 2: Lines 18-19 -This study demonstrated that serum cytokines in dogs with pancreatitis differed from those in normal controls. Therefore, these cytokines may be involved in the pathogenesis of systemic inflammation (?).

Response: As your comment, acute pancreatitis can develop into a severe form that can lead to systemic inflammatory response syndrome (We mentioned this in the introduction part: Line 45-46). Eventually, in all the patients, which suffer from acute pancreatitis, systemic inflammatory response syndrome can be developed. However, in this study, we excluded this condition by clinical signs and blood test (excluding thrombocytopenia, hypoalbuminemia and so on) and ultrasonography.

Point 3: The conclusion of the abstract is different from the conclusion at the end of the work

Response: Sorry about the confusion. We correct the conclusions in the abstract part and in the introduction part (Line 28-33, Line 74-76).

Discussion

Point 1.  Line 151 This report is one effort to elucidate the pathophysiology of acute pancreatitis in dogs. The objective needs to be more specific and determined at the beginning of the paper. My suggestion is to discuss first the results of cytokines that increased and then those that did not show up.

Response: As your comment we added the more specific goal. “This report is one effort to elucidate the pathophysiology of suspected acute pancreatitis in dogs by suggesting newly factors, MCP-1, RAGE and SCF.” (Line 174-175) And also we changed to discuss first the results of cytokines that increased and then those that did not show up as your comment (Line 225-245).

 Conclusion

Point 1: Line 119-220 - Based on the results from both assays, MCP-1, RAGE and SCF are newly suggested factors for identification of acute pancreatitis in dogs. Was this final conclusion the purpose of the work?

Response: Yes, there is few studies of MCP-1, RAGE and SCF in pancreatitis in veterinary medicine. We tried to suggest newly factors which can be candidate in pancreatitis with a meaningful role. 

Point 2: There is some recent bibliography on pancreatitis in dogs that can be cited in the paper.

Response: We added three more recent bibliographies which are related in this paper. Thank you for your suggestion

Thank you so much for your thoughtful advice. We hope this revised manuscript and letter meet your expectation.

Reviewer 2 Report

 This is a simple and well written paper. But there are some problems and it is not enough to submit as a manuscript.

 This manuscript is short and seems to be a preliminary study from the content. When considered as a preliminary study, the following two are major problems.

line 62~: You described the purpose of this study as "This study demonstrated that cytokines involved in the pathogenesis of acute pancreatitis in dogs." However, at line 220, you described the conclusions as "MCP-1, RAGE and SCF are newly suggested factors for identification of acute pancreatitis in dogs." Is it elucidation of pathophysiology or application to diagnosis? The concept of the manuscript has not been decided.

line 75~: As the author yourself described, acute pancreatitis should be diagnosed only histopathologically. The diagnosis needs to be confirmed, at least in preliminary studies. Have you performed histopathological examination in all cases?

Author Response

This is a simple and well written paper. But there are some problems and it is not enough to submit as a manuscript. This manuscript is short and seems to be a preliminary study from the content. When considered as a preliminary study, the following two are major problems.

Point 1: line 62~: You described the purpose of this study as "This study demonstrated that cytokines involved in the pathogenesis of acute pancreatitis in dogs." However, at line 220, you described the conclusions as "MCP-1, RAGE and SCF are newly suggested factors for identification of acute pancreatitis in dogs." Is it elucidation of pathophysiology or application to diagnosis? The concept of the manuscript has not been decided.

Response: Sorry about the confusion. We tried to suggest MCP-1, RAGE and SCF as newly factors for identification of acute pancreatitis in dogs. We correct all of this in the manuscript (Line 32 - 33, 74 - 76, 174 – 175, 259 - 265).

Point 2: line 75~: As the author yourself described, acute pancreatitis should be diagnosed only histopathologically. The diagnosis needs to be confirmed, at least in preliminary studies. Have you performed histopathological examination in all cases?

Response: Acute and chronic pancreatitis can only be differentiated based on histopathology. However, surgical biopsy of pancreas is invasive and requires anesthesia, which may be high risk to the patients. So surgical biopsy of pancreas is not routinely suggested to the patients. We indicated this in the manuscript (Line 50-52) So usually in clinic, the patients, which had a positive SNAP cPL test and morphologic evidence of pancreatitis obtained by ultrasonography, were considered as patients with acute pancreatitis and treated based on acute pancreatitis. As your comment we could not confirm acute pancreatitis without histological analysis so we named this as suspected acute pancreatitis. We corrected all ‘acute pancreatitis’ to ‘suspected acute pancreatitis’. Please consider all about these.

Thank you so much for your thoughtful advice. We hope this revised manuscript and letter meet your expectation. 

Reviewer 3 Report

This study described the Serum concentration of Inflammatory cytokines in dogs with Acute pancreatitis. The rational behind the experiment was clear and straight forward. The manuscript is almost well written. 

While many different sources are used to set up the study in the introduction, little previous evidence is stated. The introduction is thus short and poorly sets up the rationale for the study. More attention to how this study fits into previous work in acute pancreatitis and inflammation should be added to improve this section.

Please refer to doi: 10.3390/antiox9090781,  10.3390/antiox9100992.

Author Response

This study described the Serum concentration of Inflammatory cytokines in dogs with Acute pancreatitis. The rationale behind the experiment was clear and straight forward. The manuscript is almost well written. 

Point 1: While many different sources are used to set up the study in the introduction, little previous evidence is stated. The introduction is thus short and poorly sets up the rationale for the study. More attention to how this study fits into previous work in acute pancreatitis and inflammation should be added to improve this section.

Response: As your comment, we added more previous evidence to improve the introduction part. We described about the biopsy and ultrasonography of pancreas (Line 50-52, 66-68). Thank you for your suggestions.

Thank you so much for your thoughtful advice. We hope this revised manuscript and letter meet your expectation. 

Round 2

Reviewer 2 Report

I have no comments.